# Predictions of Aeroengines’ Infrared Radiation Characteristics Based on HKELM Optimized by the Improved Dung Beetle Optimizer

**DOI:** 10.3390/s24061734

**Published:** 2024-03-07

**Authors:** Lei Qiao, Lihai Chen, Yiwen Li, Weizhuo Hua, Ping Wang, You Cui

**Affiliations:** 1Hebei Instrument & Meter Engineering Technology Research Center, Hebei Petroleum University of Technology, Chengde 067000, China; qiaolei19841001@126.com (L.Q.);; 2Beijing Stealth Technology Co., Ltd., Beijing 100083, China; 3Key Laboratory for National Defense Science and Technology on Plasma Dynamics, Air Force Engineering University, Xi’an 710038, China

**Keywords:** aeroengine, infrared radiation characteristics, hybrid kernel extreme learning machine, dung beetle optimizer, Levy flight strategy, variable spiral strategy

## Abstract

To solve the problems of high computational cost and the long time required by the simulation and calculation of aeroengines’ exhaust systems, a method of predicting the characteristics of infrared radiation based on the hybrid kernel extreme learning machine (HKELM) optimized by the improved dung beetle optimizer (IDBO) was proposed. Firstly, the Levy flight strategy and variable spiral strategy were introduced to improve the optimization performance of the dung beetle optimizer (DBO) algorithm. Secondly, the superiority of IDBO algorithm was verified by using 23 benchmark functions. In addition, the Wilcoxon signed-rank test was applied to evaluate the experimental results, which proved the superiority of the IDBO algorithm over other current prominent metaheuristic algorithms. Finally, the hyperparameters of HKELM were optimized by the IDBO algorithm, and the IDBO-HKELM model was applied to the prediction of characteristics of infrared radiation of a typical axisymmetric nozzle. The results showed that the RMSE and MAE of the IDBO-HKELM model were 20.64 and 8.83, respectively, which verified the high accuracy and feasibility of the proposed method for predictions of aeroengines’ infrared radiation characteristics.

## 1. Introduction

In the primary stage of aircraft design, it is of great value to explore a fast and accurate prediction method for the infrared radiation features for the design of infrared stealth aircraft. The emissions contributed by the engine’s exhaust system in the infrared band of 3 to 5 μm account for more than 90% of the entire aircraft. It is one of the important problems that must be solved to carry out research on infrared stealth technology for engine exhaust systems and realize the reduction of the characteristics of infrared radiation of exhaust systems [1].

With the rapid development of computer technology and computational mathematics, a series of numerical methods have been developed to calculate the transmission of infrared radiation energy in space, such as the region method [2], the discrete coordinate method [3], the finite volume method [4], the discrete transfer method [5], the Monte Carlo method [6] and the reverse Monte Carlo method (RMCM) [7]. The intensity of infrared radiation can be obtained with high precision by numerical simulation, but it has the defects of large calculation costs and a long calculation time, which greatly reduce the efficiency of infrared stealth designs. In recent years, big data and artificial intelligence technologies have developed rapidly, and data-driven modeling methods have been widely used in the aerospace field [8]. The surrogate-based model was applied to the aerodynamic design of supercritical wings [9]. The Gaussian process model was used to improve the lift–drag ratio of airfoils [10]. A support vector machine model was built to achieve accurate and rapid predictions of rockets’ aerodynamic performance [11]. Research with machine learning models in the field of aerodynamic optimization designs and predictions of performance has been more in-depth. However, there are no reports on predictions of the characteristics of infrared radiation of aeroengines using machine learning methods.

The extreme learning machine (ELM) is a single hidden layer feedforward neural network with the traits of simplicity and quick learning [12]. However, the hidden layer of the ELM’s input weights and threshold are determined at random, making it challenging to establish the right number of hidden layers [13]. The kernel extreme learning machine (KELM) uses kernel mapping rather than random mapping, which significantly decreases the network’s complexity and improves the model’s capacity for prediction and generalization [14]. However, the KELM typically uses a single kernel function during the application process, making it challenging to adjust to samples with various data properties.

By weighting various kernel functions, the hybrid kernel extreme learning machine (HKELM) was created, which may address the issue that a single kernel function in a KELM makes it challenging to retrieve the properties of a multidimensional sample [15,16]. Additionally, the improved dung beetle optimizer (IDBO) can be applied to optimize the hybrid kernel function’s parameters. Then the IDBO-HKELM can be applied to predict the intensity of the infrared radiation of a typical axisymmetric nozzle, and the radiation intensity curve can be constructed. It was found that the IDBO-HKELM model had higher predictive performance than the other currently prominent machine learning methods for predicting the characteristics of infrared radiation. The remainder of this article is organized as follows. Section 2 presents the principles and modeling of IDBO. Section 3 presents the superiority of IDBO by using the benchmark functions. Section 4 presents the practical application and results of the analysis by using the infrared simulation dataset of a typical axisymmetric nozzle. Finally, the article is concluded in Section 5.

## 2. Principles and Modeling

### 2.1. Principle of the Hybrid Kernel Extreme Learning Machine

The extreme learning machine (ELM) is a fast single hidden layer feedforward neural network proposed in 2004. The output of the ELM can be expressed as
(1)$$f_{ELM}\left(x_{k}\right)=\sum_{j=1}^{m}\beta_{j}\cdot g\left(w_{j}\cdot x_{k}+b_{j}\right)=y_{k}$$
where $x_{k}={\left[\begin{array}{llll} x_{k1} & x_{k2} & \cdots & x_{kn} \end{array}\right]}^{T}$ is the input vector, $y_{k}={\left[\begin{array}{llll} y_{k1} & y_{k2} & \cdots & y_{kl} \end{array}\right]}^{T}$ is the output vector, $w_{j}=\left[\begin{array}{llll} w_{1j} & w_{2j} & \cdots & w_{nj} \end{array}\right]$ is the input weight of the node j in the hidden layer, $\beta_{j}={\left[\begin{array}{llll} \beta_{j1} & \beta_{j2} & \cdots & \beta_{jl} \end{array}\right]}^{T}$ is the output weight of the node j in the same hidden layer, $b={\left[\begin{array}{llll} b_{1} & b_{2} & \cdots & b_{m} \end{array}\right]}^{T}$ is the threshold of the hidden layer and $g\left(x\right)=\frac{1}{1+e^{-x}}$ is the excitation function of the hidden layer.

The output matrix $f_{ELM}\left(x_{k}\right)$ can be written as $f_{ELM}\left(X\right)=H\beta =Y$, where H is the output matrix of the hidden layer, which can be expressed as:(2)$$H={\left[\begin{array}{ccc} g\left(w_{1}\cdot x_{1}+b_{1}\right) & \cdots & g\left(w_{m}\cdot x_{1}+b_{m}\right)\\ g\left(w_{1}\cdot x_{2}+b_{1}\right) & \cdots & g\left(w_{m}\cdot x_{2}+b_{m}\right)\\ \vdots & \vdots & \vdots \\ g\left(w_{1}\cdot x_{N}+b_{1}\right) & \cdots & g\left(w_{m}\cdot x_{N}+b_{m}\right) \end{array}\right]}_{N\times m}$$

The input weight matrix W and the threshold b can be randomly given. The output weight matrix β is calculated by the formula $\beta =H^{+}\cdot Y$, where $H^{+}$ is the pseudo-inverse matrix of H.

The ELM overcomes the defect of the traditional neural networks, including the slow training speed, easy overfitting and local extreme values, but it still has some shortcomings, such as the difficulty of determining the number of nodes in the hidden layer, and the possibility that the output matrix of the hidden layer is not satisfied with the rank, which leads to the existence of ill-conditioned solutions. β is obtained via the pseudo-inverse without regularization, which leads to easy overfitting. In view of the shortcomings of ELM, inspired by the introduction of the kernel function in the support vector machine, the kernel extreme learning machine (KELM) was proposed. The kernel matrix is defined as
(3)$$\Omega_{KELM}\left(X\right)=h\left(X\right)\cdot H^{T}$$
where $h\left(X\right)$ is the mapping from the input to the output of the hidden layer.

The elements at row i and column j of the matrix are
(4)$$\Omega_{{KELM}_{ij}}\left(x_{i},x_{j}\right)=h\left(x_{i}\right)\cdot h\left(x_{j}\right)≜K\left(x_{i}\cdot x_{j}\right)$$
where $K\left(x_{i}\cdot x_{j}\right)$ is the kernel function. In addition $f_{KELM}\left(X\right)$ can be estimated by Formula (5), which is described below:(5)$$f_{KELM}\left(X\right)=\left[K\left(X\cdot x_{1}\right)\;\cdots K\left(X\cdot x_{N}\right)\right]\times {\left(\Omega_{KELM}\left(X\right)\right)}^{-1}\cdot Y$$

After introducing the regularized item:(6)$$f_{KELM}\left(X\right)=\left[K\left(X\cdot x_{1}\right)\;\cdots \;K\left(X\cdot x_{N}\right)\right]\times {\left(\frac{I}{C}+\Omega_{KELM}\left(X\right)\right)}^{-1}\cdot Y$$

The network output $f_{KELM}\left(X_{k}\right)$ of the KELM can be written as
(7)$$f_{KELM}\left(X_{k}\right)=\sum_{k=1}^{n}\Omega_{KELM}\left(X\cdot x_{k}\right)\cdot {\left((\frac{I}{C}{)}_{k}+\Omega_{KELM}\left(X\cdot x_{k}\right)\right)}^{-1}\cdot Y_{k}$$
where *C* is the regular term coefficient. The accuracy of the model increases as *C* increases, but overfitting is more likely to happen. Conversely, as *C* decreases, the generalization ability increases and the error-tolerant rate increases, but underfitting is more likely to happen. Therefore, an appropriate C value is crucial for the model.

The KELM significantly reduces the network’s complexity by using a kernel function to translate a low-dimensional space to a high-dimensional space, improving the prediction and generalization ability. The typical KELM algorithm’s single kernel function, however, struggles to handle a variety of sample data. This led to the suggestion of a hybrid kernel extreme learning machine (HKELM). The flaw of the single-core ELM can be fixed, and the issues of poor generalization and low prediction accuracy can be resolved by incorporating a hybrid kernel.

There are many types of kernel functions, among which, the radial basis function is a typical local kernel function with strong learning ability but weak generalization ability, and the polynomial function is a typical global kernel function with weak learning ability but strong generalization ability. To further improve the learning and generalization performance of the kernel extreme learning machine, the radial basis function and polynomial function were chosen to carry out a weighted combination, and the equivalent kernel function combining the two kinds of kernels was constructed.

The polynomial kernel function is
(8)$$K_{\mathrm{poly}}\left(X,x_{k}\right)={\left[\left(x_{k}^{T}X\right)+C_{0}\right]}^{d},C_{0}>0$$
where $C_{0}$ and d are the parameters of the polynomial kernel function.

The polynomial kernel function is a typical global kernel function. The curve of the polynomial kernel function is shown in Figure 1, where the test points x=0.5, $C_{0}=1$ and d are set to 1, 2, 3 and 4, respectively. As shown in Figure 1, sample points that are far away from the test point x=0.5 can still have a great impact on the value of the kernel function, so it is easy to extract the overall characteristics of the sample’s information, and the model has excellent generalization ability.

The radial basis kernel function is
(9)$$K_{RBF}\left(X,x_{k}\right)=exp\left(\frac{-{∥X-x_{k}∥}^{2}}{2\sigma^{2}}\right)$$
where *σ* is the kernel’s width.

The radial basis kernel function is a typical local kernel function with excellent local interpolation ability. The curve of the radial basis kernel function is shown in Figure 2, with the test point x=0.5, and σ is set to 0.1, 0.2, 0.3 and 0.4. The value of the kernel function in the area near the test point x=0.5 is not 0, and the value of the kernel function rapidly approaches 0 with an increase in the distance. Therefore, it is convenient to extract the local characteristics of the samples, and the model has excellent learning ability.

The equivalent kernel function of HKELM can be expressed as
(10)$$\Omega_{HKELM}\left(X,x_{k}\right)=C_{1}K_{\mathrm{poly}}\left(X,x_{k}\right)+C_{2}K_{RBF}\left(X,x_{k}\right)$$
where $C_{1}$ and $C_{2}$ are the weighted coefficients of the kernel function, ranging across [0, 1], and $C_{1}+C_{2}=1$. After cancelling out $C_{2}$, we have
(11)$$\Omega_{HKELM}\left(X,x_{k}\right)=C_{1}\cdot K_{\mathrm{poly}}\left(X,x_{k}\right)+\left(1-C_{1}\right)\cdot K_{RBF}\left(X,x_{k}\right)$$
where *σ* is the kernel’s width.

The hybrid kernel function curve is shown in Figure 3, with the test point x=0.5, and $C_{1}$ is set to 0.1, 0.2, 0.3 and 0.4. In $K_{\mathrm{poly}}\left(X,x_{k}\right)$, σ is set to 0.1. In $K_{RBF}\left(X,x_{k}\right)$, m is set to 1, n is set to 1 and d is set to 2. It can be seen from Figure 3 that the hybrid kernel function not only affects the sample points around the test point, but also the sample points with a certain distance from the test point. Therefore, the hybrid kernel function effectively combines the advantages of the radial basis kernel function and the polynomial kernel function, making up for the shortcomings of the single-kernel function.

The output of HKELM can be expressed as:(12)$$f_{HKELM}\left(x_{k}\right)=\sum_{k=1}^{N}\Omega_{HKELM}\left(X\cdot x_{k}\right)\left\{C_{1}{\left[\left(x_{k}^{T}X\right)+C_{0}\right]}^{d}+\right.\left(1-C_{1}\right)\;{\left.exp\left(\frac{{∥X-x_{k}∥}^{2}}{2\sigma^{2}}\right)+{\left(\frac{I}{C}\right)}_{k}\right\}}^{-1}\cdot Y_{k}$$

The hybrid kernel function of HKELM combines the advantages of the global kernel and the local kernel. As a result, the HKELM not only enhances the global search capabilities but also has good local search capabilities. The hyperparameters ($C_{0},C_{1},\;d,\sigma ,C$) of the hybrid kernel function should be optimized, since the hybrid kernel function has many parameters which are inefficiently to determine manually.

### 2.2. Principle of the Dung Beetle Optimizer

The dung beetle optimizer (DBO) is a swarm intelligence optimization algorithm proposed by Jiankai Xue in 2022 [17]. It was inspired by the ball-rolling, dancing, foraging, stealing and reproduction behaviors of beetles and has the characteristics of strong optimization ability and a fast convergence speed.

The position of the ball-rolling dung beetle can be described as
(13)$$\left\{\begin{array}{c} x_{i\;}\left(t+1\right)=x_{i\;}\left(t\right)+\alpha \times k\times x_{i\;}\left(t-1\right)+b\times ∆x\\ ∆x=\left|x_{i\;}\left(t\right)-X^{w}\right|\; \end{array}\right.$$
where t is the current iteration number, $x_{i\;}\left(t\right)$ is the information about the ith dung beetle’s position at the tth iteration, k is a constant value that represents the deflection coefficient, b is a constant value in the range of (0, 1), α is a natural coefficient that is given a value of either −1 or 1, $X^{w}$ is the global worst position and ∆x is used to simulate changes in light intensity. Choosing the right values for the two parameters (k and b) is quite important. Take note that α indicates that dung beetles can be diverted from their intended path by a variety of environmental elements, including wind and uneven terrain. To be more precise, α=1 denotes no deviation and α=−1 denotes a departure from the initial direction. To mimic a complicated environment, the probability technique is used to fix α to either 1 or −1. Likewise, a larger value of ∆x indicates a weaker light source. Furthermore, k and b are set to 0.1 and 0.3, respectively. The ∆x can encourage the ball-rolling dung beetle to pursue the following two benefits: (1) during the optimization process, extensively investigate the entire issue space; (2) pursue stronger searching performance and lower the likelihood of slipping into the local optima.

When the dung beetle encounters an obstacle and is unable to move forward, it needs to dance to reposition itself, and once the beetle has successfully determined a new direction, it continues to push the ball. The position of the beetle is updated and defined as follows
(14)$$x_{i\;}\left(t+1\right)=x_{i\;}\left(t\right)+tan(\theta )\left|x_{i\;}\left(t\right)-x_{i\;}\left(t-1\right)\right|$$
where θ is the deflection angle in the range of (0,π). Using the tangent function of θ, we obtain the new rolling direction and can imitate the dancing behavior.

In the wild, dung beetles roll dung balls to safety and conceal them. The boundary selection strategy used to simulate the spawning region where female beetles lay the eggs is described by
(15)$$\left\{\begin{array}{c} {Lb}^{*}=max\;(X^{*}\times \left(1-R\right),Lb)\\ {Ub}^{*}=min\;(X^{*}\times \left(1+R\right),Ub) \end{array}\right.$$
where $X^{*}$ is the current local best location; ${Lb}^{*}$ and ${Ub}^{*}$ are the lower and upper bounds of the spawning area, respectively; $R=1-t/T_{max}$ and $T_{max}$ is the maximum number of iterations. Lb and Ub are the lower and upper limits of the optimization issue, respectively. It can be clearly seen that the boundary range of the spawning area changes dynamically, which is mainly determined by R. As t gradually increases, R decreases linearly, and the spawning area gradually shrinks. This can reduce the chance of oscillating back and forth when convergence is near the optimal point.

The female beetles select the brood balls in this region to lay eggs once the spawning location has been determined. In the iteration phase, the brood ball’s position is dynamic and can be described as
(16)$$B_{i\;}\left(t+1\right)=X^{*}+b_{1}\times \left(B_{i}\left(t\right)-{Lb}^{*}\right)+b_{2}\times \left(B_{i}\left(t\right)-{Ub}^{*}\right)$$
where $B_{i}\left(t\right)$ is the position of the *i*th brood ball at the tth iteration, and $b_{1}$ and $b_{2}$ are the two independent random vectors. The random vectors $b_{1}$ and $b_{2}$ can effectively reduce the risk of falling into the local optimal solution by adding random elements to the motion.

The boundary of the optimal foraging region for small dung beetles is defined as follows
(17)$$\left\{\begin{array}{c} {Lb}^{b}=max\;(X^{b}\times \left(1-R\right),Lb)\\ {Ub}^{b}=min\;(X^{b}\times \left(1+R\right),Ub) \end{array}\right.$$
where $X^{b}$ is the global best position, and ${Lb}^{b}$ and ${Ub}^{b}$ are the lower and upper limits of the optimal foraging region, respectively.

The position of the small dung beetle can be revised as follows
(18)$$x_{i\;}\left(t+1\right)=x_{i\;}\left(t\right)+C_{1}\times \left(x_{i}\left(t\right)-{Lb}^{b}\right)+C_{2}\times \left(x_{i}\left(t\right)-{Ub}^{b}\right)$$
where $x_{i\;}\left(t\right)$ is the position of the ith small dung beetle at the tth iteration, $C_{1}$ is a random number with normally distributed and $C_{2}$ is a random vector between 0 and 1.

Some dung beetles are referred to be thieves because they take other dung beetles’ dung balls. The position of the thief can be described as follows:(19)$$x_{i\;}\left(t+1\right)=X^{b}+S\times g\times (\left|x_{i\;}\left(t\right)-X^{*}\right|+\left|x_{i\;}\left(t\right)-X^{b}\right|)$$
where $x_{i\;}\left(t\right)$ is the position of the ith thief at the tth iteration, g is a random vector with a normally distribution and S is a constant value. The introduction of g increases the random disturbance and can avoid falling into the local optimal solution to a certain extent. S can be used to update the stride size of the iteration. For more information on dung beetle optimization, please refer to the related article [12].

### 2.3. Improvement of the DBO

The DBO algorithm has the characteristics of few adjustment parameters, strong optimization ability and a fast convergence speed. But it easily falls into the local optimal, and the global exploration ability is weak. The DBO algorithm should be optimized to improve the global optimization capability.

#### 2.3.1. Levy Flight Strategy

The Levy flight strategy was proposed to improve the DBO algorithm. Levy flight is a random search based on Levy distribution, which has been cited many times in the optimization field and is often used to improve swarm intelligence optimization algorithms such as the sparrow search algorithm (SSA) [18], the whale optimization algorithm (WOA) [19] and particle swarm optimization (PSO) [20]. The Levy flight strategy enables the algorithm to change randomly between short and long distances, uses a few long hops to avoid the algorithm falling into local optima, and enhances the global optimal search ability.

The location of the thief can be updated according to Equation (20)
(20)$$x_{i\;}\left(t+1\right)={Levy(\lambda )\times X}^{b}+S\times g\times (\left|x_{i}\left(t\right)-X^{*}\right|+\left|x_{i}\left(t\right)-X^{b}\right|)$$
where Levy(λ) indicates that it follows the Levy distribution with the parameter λ. It can be expressed as:(21)$$Levy\left(\lambda \right)\sim \mu =t^{\mu }$$

The Mantegna algorithm can be used to simulate Levy flight, and its mathematical expression is as follows
(22)$$s=\frac{\mu }{{\left|\nu \right|}^{\frac{1}{\beta }}}$$
where s is the Levy flight path, μ and ν are the random numbers with a normal distribution, and $\mu \sim N(0,\sigma_{\mu }^{2})$, $\nu \sim N(0,\sigma_{\nu }^{2})$. $\sigma_{\mu }$ and $\sigma_{\nu }$ are obtained by Equation (23):(23)$$\left\{\begin{array}{c} \sigma_{\mu }={\left\{\frac{\Gamma (1+\beta )sin\;(\pi \beta /2)}{\Gamma \left[\frac{\left(1+\beta \right)}{2}\right]\beta 2^{\frac{\left(\beta -1\right)}{2}}}\right\}}^{1/\beta }\\ \sigma_{\nu }=1 \end{array}\right.$$

The value of parameter β ranges from 0 to 2, and, in general, β = 1.5.

#### 2.3.2. Variable Spiral Strategy

Inspired by the whale optimization algorithm, the variable spiral strategy was introduced to improve the position of the brood ball and the small dung beetle. The whale spiral search model is shown below
(24)$$\left\{\begin{array}{c} X_{t+1}={\hat{X}}_{t}+D\cdot e^{bl}\cos(2\pi l)\\ D=\left|2r\cdot {\hat{X}}_{t}-X_{t}\right|\; \end{array}\right.$$
where $X_{t+1}$ is the position of the whale when iteratively searching t+1 times, ${\hat{X}}_{t}$ is the global optimal position, b is a constant that determines the shape of the spiral, and l and r are uniformly distributed random numbers on the interval [0, 1]. The spiral search can be applied to update the position of the brood ball and the small dung beetle.

In each spiral search, whales approach their prey with a fixed spiral path, which easily makes them fall into local optima. Therefore, the spiral constant b is modified as a parameter varying with the number of iterations, and the spiral’s shape is dynamically adjusted to broaden the global search area of the swarm, thus improving the global search ability. The model of the variable spiral parameter m is as follows:(25)$$m=e^{5cos\;(\pi (1-\frac{t}{t_{max}}))}$$

As can be seen from Equation (24), the spiral’s shape decreases as the number of iterations increases. At the beginning, the brood ball and the small dung beetle conduct a global search in the largest possible area and gradually approach the optimal solution in the later iterations, which improves the global search ability and optimization accuracy of the algorithm.

The position of the brood ball can be updated according to Equation (26):(26)$$B_{i\;}\left(t+1\right)=X^{*}+e^{ml}cos\;(2\pi l)\times \left(B_{i}\left(t\right)-{Lb}^{*}\right)+e^{ml}cos\;(2\pi l)\times \left(B_{i}\left(t\right)-{Ub}^{*}\right)$$

The position of the small dung beetle can be updated according to Equation (27):(27)$$x_{i\;}\left(t+1\right)={e^{ml}cos\;(2\pi l)\times x}_{i\;}\left(t\right)+C_{1}\times \left(x_{i}\left(t\right)-{Lb}^{b}\right)+C_{2}\times \left(x_{i}\left(t\right)-{Ub}^{b}\right)$$

The calculation process of the IDBO is as follows:(1)Initialize the IDBO algorithm’s settings and the dung beetle swarm.(2)Calculate the fitness values of all agents according to the objective function.(3)Update the position of the ball-rolling dung beetle by using Equations (13) and (14), the position of the thief by using Equation (20), the position of brood ball by using Equation (26), and the position of small dung beetle by using Equation (27).(4)Determine whether each agent is outside the limit.(5)Reevaluate the fitness value of the current optimal solution.(6)Keep repeating the previous stages until it satisfies the termination requirement.

## 3. IDBO Algorithm Performance Test

### 3.1. Test Functions and Parameter Settings

In this section, 23 classical test functions were used to analyze and verify the improvement effect of the IDBO algorithm. The 23 reference functions can be roughly divided into three categories: the unimodal (F1–F7), the multimodal (F8–F13) and the fixed-dimension multimodal functions (F14–F23). To further verify the specific effect after the improvement, the IDBO algorithm was compared with other heuristic algorithms (the DBO, SSA, WOA and PSO). The population size of all algorithms was set to 50, and the number of population iterations was 500. It is important to note that each simulation experiment was run independently 30 times in an effort to eliminate random effects. The mean and standard deviation (Std Dev) of each algorithm were calculated.

### 3.2. Optimization Capability of IDBO (Functions F1–F23)

The specific information about the simulation results is shown in Table 1. The Std Dev and the mean fitness value are displayed to gauge how well the algorithms conducted searches.

The unimodal test functions (F1–F7) have only one global best solution, which can be used to test the exploitation performance. It can be observed from Table 1 that the IDBO algorithm outperformed the other algorithms for the test functions F1–F5, as measured by the evaluation indicators, including the mean and Std Dev. For functions F6–F7, the fitness value of IDBO algorithm was not the best, but it was also ranked in the front.

The multimodal functions F8–F23 can be used to verify the exploration ability. The functions F8–F13 have many local minimum values which grow exponentially as the dimensions increase. For the functions F8 and F12–F13, the IDBO algorithm achieved greater search ability than the other algorithms proposed. For the functions F9–F11, the IDBO algorithm was the same as DBO and SSA, and performed better than the WOA and PSO. The functions F14–F23 are the multimodal functions with fixed dimensions. The IDBO algorithm achieved a superior search performance than the other algorithms for the functions F14–F15, and F19–F23. For the functions F16–F18, the search capabilities of all algorithms were similar, and the optimization results obtained by the IDBO algorithm were competitive.

### 3.3. Comparative Analysis of the Algorithms’ Convergence Curves

To show the convergence effect of each algorithm in test functions, the corresponding algorithm convergence curves were drawn according to the generated data. Figure 4 shows the fitness curves of IDBO and other algorithms in the optimization process of the partial benchmark functions. It can be clearly seen that the IDBO was superior to other algorithms in terms of the optimization speed and convergence accuracy.

### 3.4. Statistical Analysis: Rank Sum Test

To further compare the differences between the IDBO and other optimization algorithms, a statistical test called the Wilcoxon signed-rank test was conducted [21]. The significance of the statistical results were determined by calculating the *p*-value. If the *p*-value was <0.05, it was concluded that there was a significant difference between the two algorithms. The results of the calculation are shown in Table 2. It can be seen from Table 2 that the IDBO’s search results were less similar to those of its competitors. Therefore, the optimization performance of the proposed IDBO on the 23 benchmark functions was significantly different from that of other metaheuristic algorithms. Combined with the analysis in this section, it can be seen that the IDBO’s comprehensive performance was the most outstanding among many metaheuristic algorithms.

## 4. Practical Application and Analysis of the Results 

### 4.1. Data Preparation

A simplified geometric model of the nozzle is shown in Figure 5. The model consisted of an inner passage, an outer culvert, a center cone, a lobe mixer, a flame stabilizer, an afterspray pipe, a heat shield, an afterburner, a contraction section, an expansion section and an adjusting plate.

#### 4.1.1. Flow Field Calculation

The calculation domain of the flow field included two parts: inside the nozzle and outside the nozzle. The outflow field was a long straight cylinder starting from the inlet of the regulator and extending backward along the nozzle. The front of the engine nozzle was placed inside the aircraft, shielded by the fuselage, and the exposed part was behind the regulator. The flow field’s calculation domain is shown in Figure 6. The nozzle’s diameter was set as D, and the specific size of the outflow field was set with an axial length of 42D and a radial size of 12D.

The flow characteristics of the nozzle can be captured more accurately by the structured mesh division of the nozzle mode. The distribution of grids on the symmetric plane after verification of the independence is shown in Figure 7, and the total number of grids was about 5 million.

The grid divisions of each component of the exhaust are shown in Figure 8, and the total number of grids was about 3 million.

The turbulence model (sst k−ε) was used to calculate the flow field, and the discrete coordinate radiation model (DO model) was added to consider the influence of radiation heat transfer. The standard of residual convergence was set to be less than $1\times 10^{-4}$.

#### 4.1.2. Calculation of the Characteristics of Infrared Radiation 

The infrared spectral radiation intensity (ISRI) of the nozzle in the waveband of 3 to 5 μm was simulated by using the RMCM. The infrared integrated radiation intensity (IIRI) can be obtained by integrating the ISRI in the band of 3 to 5 μm. The schematic diagram of the detection of the bearings’ definition is shown in Figure 9.

The nozzle outlet’s center was taken as the origin of the coordinates, and the unit vector of the nozzle’s axis direction was taken as the Z axis. The detection points were arranged on the sphere with the radius of the detection distance, so any azimuth of detection could be expressed in the form of spherical coordinates ($\theta_{d},\varphi_{d},r_{d}$). Considering the axial symmetry of the model, only one symmetrical plane was used to arrange the detection points. The range of the elevation angle $\theta_{d}$ was 0 to 90° and the interval $\Delta \theta_{d}$ was 10°. Because the nozzle was axisymmetric, the azimuth $\varphi_{d}$ could be ignored. The detection distance $r_{d}$ was 5 to 90 km, and the distance interval $\Delta r_{d}$ was 5 km. The calculated spectral range was 3 to 5 μm. The spectral resolution Δλ was 0.05 μm. When the detection distance was 10 km, the distribution of dimensionless ISRI of the nozzle in the direction of the elevation angle from 0 to 90° is shown in Figure 7. 

It can be seen from Figure 10 that the ISRI of the prototype nozzle in the waveband of 3 to 5 μm can be roughly divided into two parts: the first part is the infrared radiation in the waveband of 3.2 to 4.2 μm, which is mainly the solid wall’s radiation. The second part is the infrared radiation in the waveband of 4.4 to 4.8 μm, which is mainly caused by the absorption and emission of carbon dioxide. As the elevation angle increases, the detectable area of the solid wall and the solid wall’s temperature increased accordingly, and the proportion of gas radiation decreased gradually.

### 4.2. IDBO-HKELM Prediction Flow

The IDBO was used to optimize five parameters of the HKELM including the polynomial kernel function parameters $C_{0}$ and d, the radial basis kernel function width σ, the kernel weighting coefficient $C_{1}$ and the the regular term coefficient C. These five tuning parameters control the prediction, learning and generalization of the HKELM. The goal of the optimization process is to compare different parameter combinations and select the optimal parameter combination with the lowest error.

Figure 11 shows the IDBO-HKELM prediction flow.

The specific steps of IDBO-HKELM are as follows.

Step 1: Divide the input dataset into a training set and test set, and normalize the data to [0, 1] using the max–min normalization method. The expression of the max–min normalization method is
(28)$$x^{\prime }=\frac{x-x_{min}}{x_{max}-x_{min}}$$
where x is the actual vector; $x_{max}$ and $x_{min}$ are the maximum and minimum values of the vector *x*, respectively; and $x^{\prime }$ is the normalized vector.

Step 2: Establish an objective function model. The objective function is the root mean square error (*RMSE*), which is expressed as
(29)$$RMSE=\sqrt{\frac{1}{N}\sum_{i=1}^{N}{\left(y_{k}-{\hat{y}}_{k}\right)}^{2}}$$
where $y_{k}$ is the true value and ${\hat{y}}_{k}$ is the predicted value of the HKELM.

Step 3: Use the IDBO is to optimize the parameters in the HKELM. The optimal individual (the optimal parameters in HKELM) is selected by judging the value of the fitness function or the maximum number of iterations.

Step 4: Apply the optimal combinations of the parameters to the estimation of the HKELM.

### 4.3. Comparative Analysis of the Models’ Prediction Effect

The two parameters ($\theta_{d}$ and $r_{d}$) of the detection point were selected as inputs, and the IIRI of the detection point was used as the object of prediction. According to the IIRI data obtained by the simulation calculation, 80% of the data were taken as the training set, 20% of the data were taken as the test set, and the machine learning models (back propagation neural network (BPNN), ELM, KELM and IDBO-HKELM) were established.

The fitness reduction rate in the training phase of the HKELM is shown in Figure 12. As can be observed, the IDBO converged more quickly than the rival algorithms, and the final error resulting from the IDBO was lower than that obtained by the rival algorithms. This further verified the superiority of the IDBO.

The specific parameters optimized by the IDBO are displayed in Table 3.

The IIRI prediction results of the IDBO-HKELM and the rival models are shown in Table 4. The predictive performance of ELM was better than that of BPNN, which showed the superiority of the ELM algorithm. The prediction accuracy of KELM was better than that of ELM, indicating that the addition of a kernel function can improve the predictive performance. KELM mapped the input samples to the high-dimensional hidden feature space via the kernel function, replaced random mapping with kernel mapping, enhanced the learning ability and generalization ability, and its fitting ability was better than that of the non-kernel extreme learning machine. The IDBO-HKELM had the lowest RMSE and MAE. Because the IDBO-HKELM model adopted a hybrid kernel function, this effectively combined the advantages of the radial basis kernel function and the polynomial kernel function. The learning and generalization ability was further improved by obtaining the optimal weights with the IDBO algorithm.

The cross-plots of IIRI calculated by RMCM versus the predicted IIRI with the BPNN, ELM, KELM and IDBO-HKELM models are shown in Figure 13. It can be concluded that the IDBO-HKELM model had the best predictive performance.

According to the method of building the IIRI model, the ISRI models (BPNN, ELM, KELM, and IDBO-HKELM) corresponding to each specific band (spectral range: 3–5 μm; spectral resolution: 0.05 μm) can be established. The infrared spectral curves can be obtained by interpolating the predicted values of each specific band. For when the distance was 10 km, the predicted infrared spectral curves are shown in Figure 14. It can be clearly observed that the shape of the infrared spectral curve predicted by the IDBO-HKELM model was closer to the curve calculated by RMCM. The IDBO-HKELM model had a good overall predictive performance on the infrared spectral curves.

## 5. Conclusions

The prediction of aeroengines’ characteristics of infrared radiation is an important part of aircraft stealth designs. To solve the problems of high computational complexity and the long calculation time of the numerical simulation method, a data-driven model based on the IDBO-HKELM was proposed. The infrared simulation dataset of an axisymmetric nozzle was used as the training and testing samples. It was found that the IDBO-HKELM model had high accuracy for the infrared spectral characteristics of different bands. Therefore, it can be confidently stated that the IDBO-HKELM model is much better suited for predicting the characteristics of infrared radiation compared with the rival models proposed in this research. The potential areas for future research include the influence of the structural and aerodynamic parameters on the characteristics of infrared radiation, using machine learning methods.

## Figures and Tables

**Figure 1 sensors-24-01734-f001:**
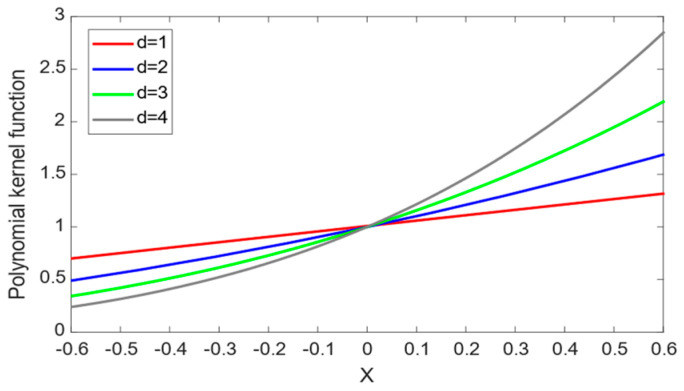
Polynomial kernel function curve.

**Figure 2 sensors-24-01734-f002:**
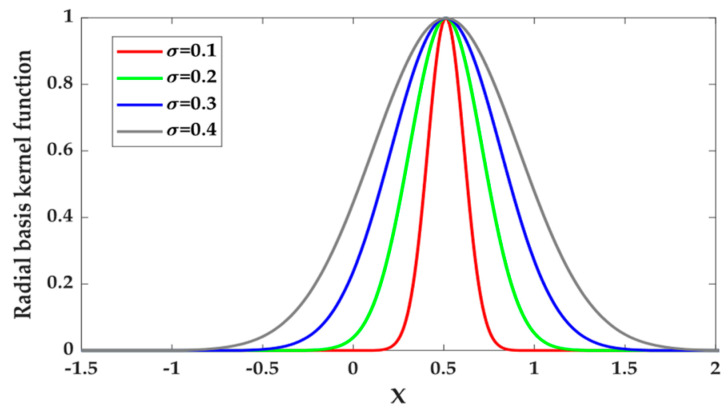
Radial basis kernel function curve.

**Figure 3 sensors-24-01734-f003:**
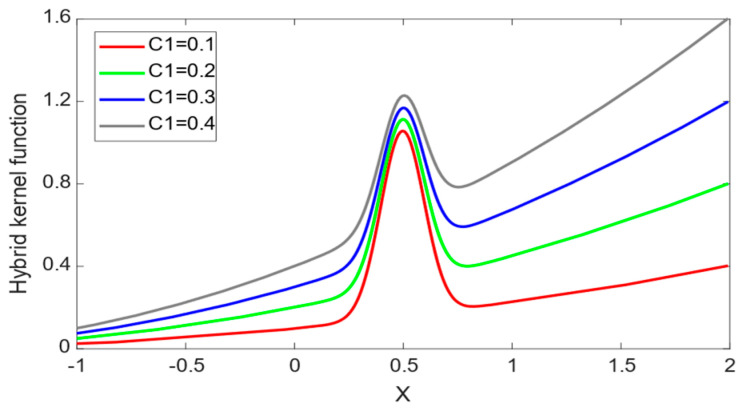
Hybrid kernel function curve.

**Figure 4 sensors-24-01734-f004:**
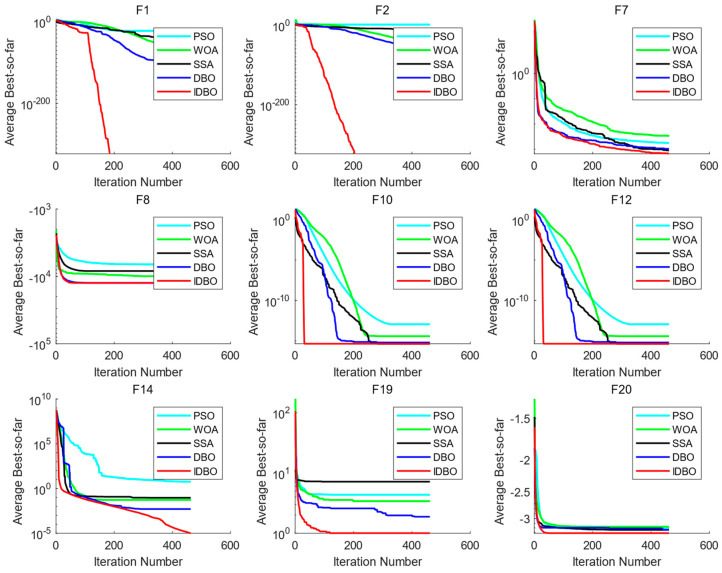
Convergence curves of the IDBO and other algorithms on some test functions.

**Figure 5 sensors-24-01734-f005:**
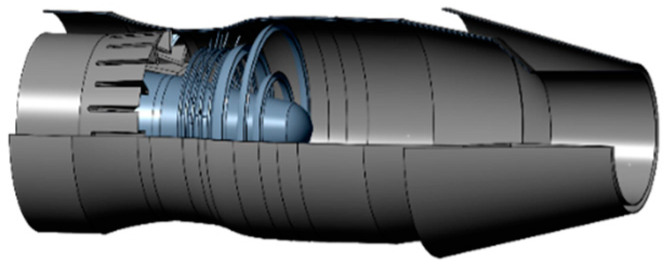
Schematic diagram of the simplified model of an axisymmetric nozzle.

**Figure 6 sensors-24-01734-f006:**
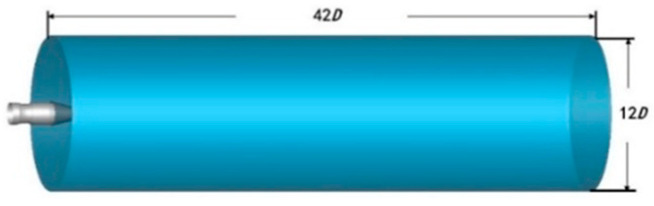
Diagram of the flow field calculation domain.

**Figure 7 sensors-24-01734-f007:**
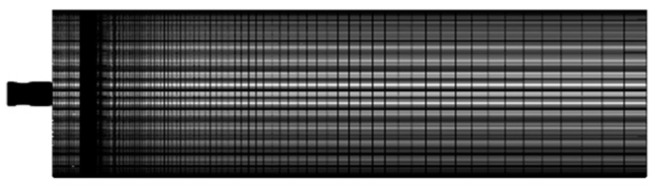
Symmetric surface grid of the flow field calculation domain.

**Figure 8 sensors-24-01734-f008:**
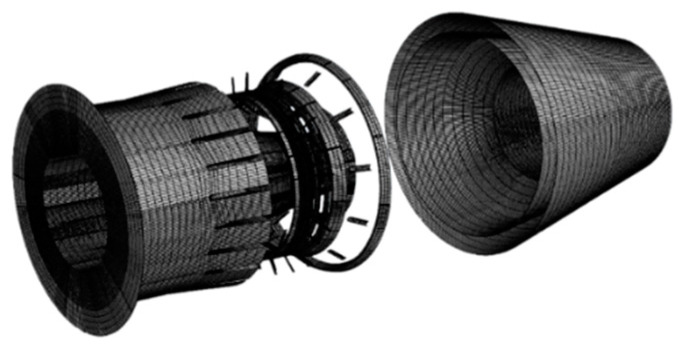
Grid of the exhaust system\s components.

**Figure 9 sensors-24-01734-f009:**
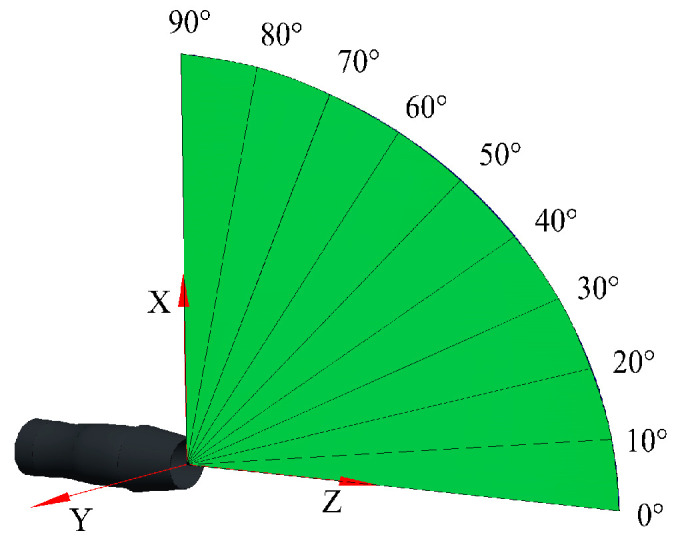
Definition of the azimuth of detection.

**Figure 10 sensors-24-01734-f010:**
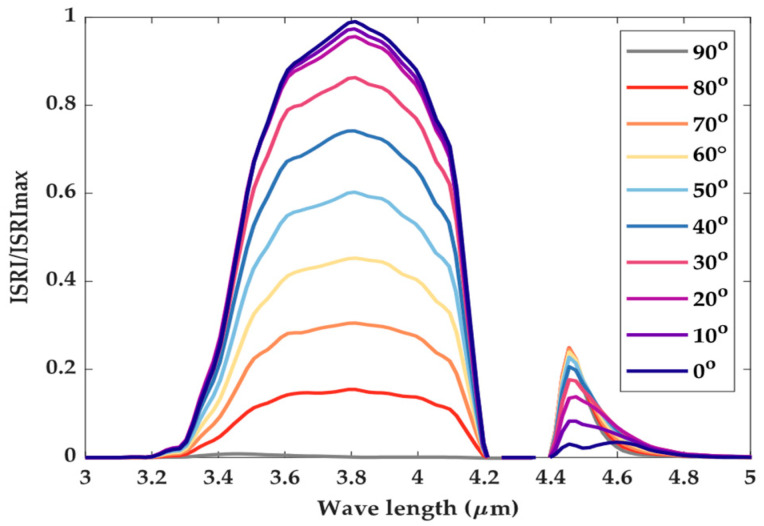
Dimensionless ISRI distribution of the axisymmetric nozzle.

**Figure 11 sensors-24-01734-f011:**
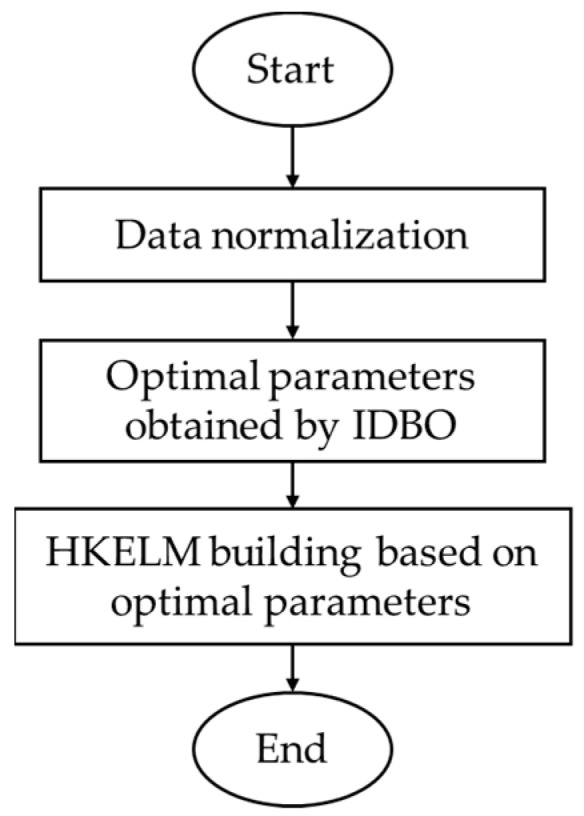
IDBO-HKELM prediction flow.

**Figure 12 sensors-24-01734-f012:**
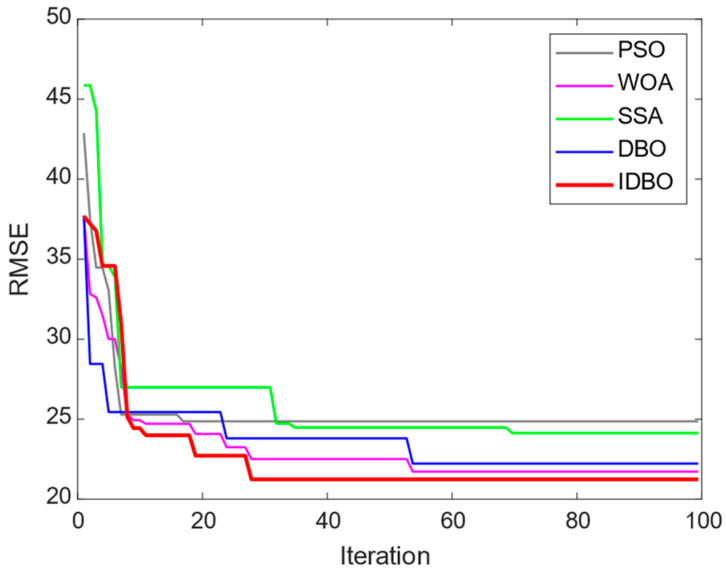
Fitness reduction rate in different iterations of the IDBO for training the HKELM.

**Figure 13 sensors-24-01734-f013:**
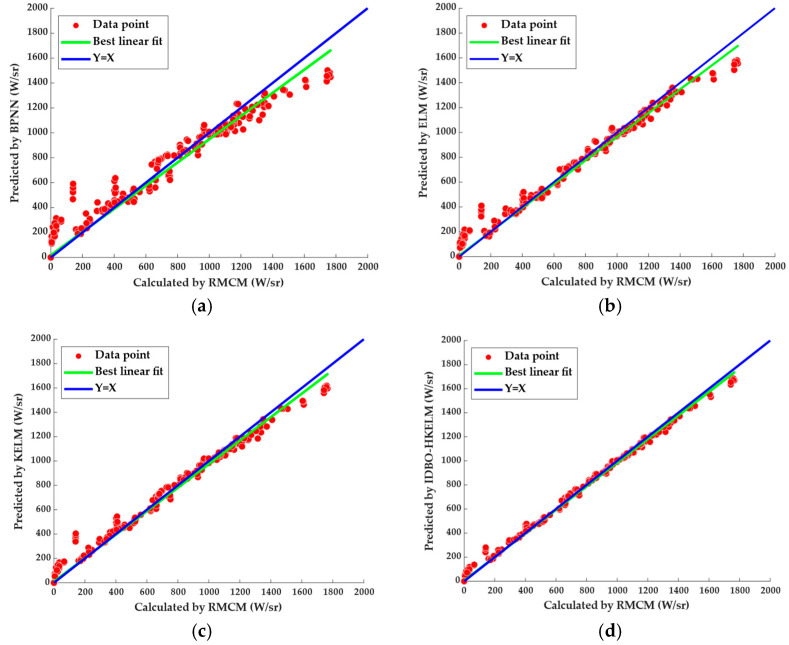
Prediction results of different models. (**a**) BPNN; (**b**) ELM; (**c**) KELM; (**d**) IDBO-HKELM.

**Figure 14 sensors-24-01734-f014:**
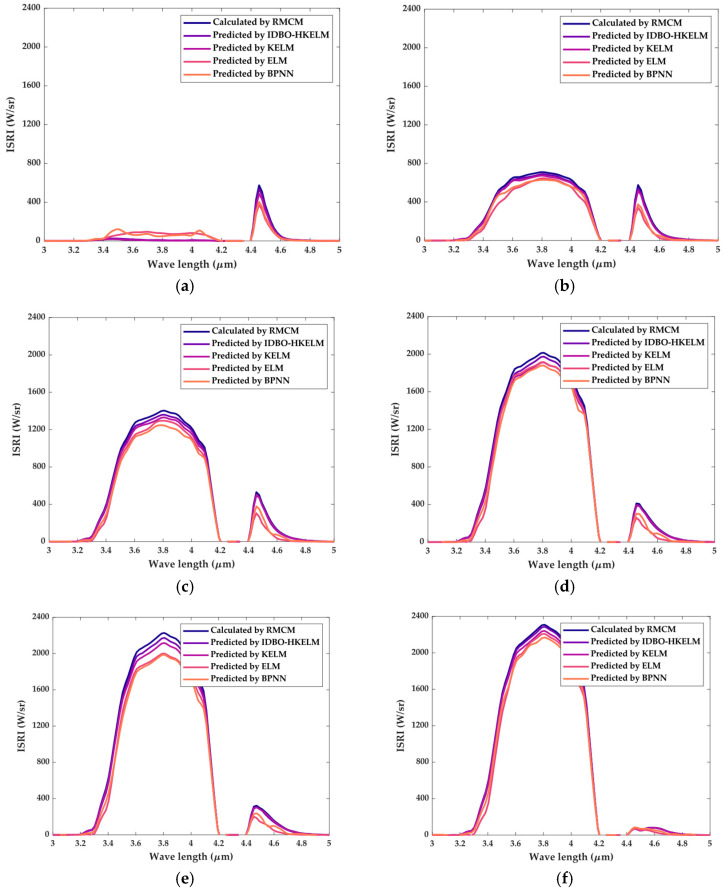
Predicted results of the infrared spectral curves of different models: (**a**) elevation angle: 90°; (**b**) elevation angle: 70°; (**c**) elevation angle: 50°; (**d**) elevation angle: 30°; (**e**) elevation angle: 20°; (**f**) elevation angle: 0°.

**Table 1 sensors-24-01734-t001:** Results of IDBO, DBO, SSA, WOA, and PSO on CEC–2022 benchmark functions. The Std Dev is given in parentheses and the best results are shown in bold.

F	IDBO	DBO	SSA	WOA	PSO
F1	**0.00**	1.67 × 10^−112^	2.24 × 10^−87^	1.35 × 10^−73^	1.96 × 10^−27^
	(0.00)	(8.79 × 10^−112^)	(4.80 × 10^−87^)	(5.02 × 10^−73^)	(5.48 × 10^−27^)
F2	**0.00**	3.41 × 10^−52^	1.35 × 10^−45^	1.35 × 10^−52^	8.03 × 10^−17^
	(0.00)	(1.86 × 10^−51^)	(1.30 × 10^−45^)	(3.38 × 10^−52^)	(5.28 × 10^−17^)
F3	**1.20 × 10^−99^**	8.82 × 10^−57^	4.57 × 10^−27^	2.08 × 10^−22^	1.03 × 10^−5^
	(6.60 × 10^−99^)	(4.83 × 10^−56^)	(2.08 × 10^−26^)	(8.61 × 10^−22^)	(2.22 × 10^−5^)
F4	**2.19 × 10^−66^**	1.00 × 10^−53^	1.22 × 10^−26^	2.73 × 10^−37^	5.26 × 10^−7^
	(1.19 × 10^−65^)	(5.50 × 10^−53^)	(6.68 × 10^−26^)	(3.32 × 10^−37^)	(4.51 × 10^−7^)
F5	**2.86 × 10^1^**	2.57 × 10^1^	2.59 × 10^1^	2.78 × 10^1^	2.71 × 10^1^
	(3.20 × 10^−1^)	(2.60 × 10^−1^)	(3.70 × 10^−1^)	(5.40 × 10^−1^)	(7.85 × 10^−1^)
F6	1.40 × 10^−2^	**1.05 × 10^−2^**	7.32 × 10^−1^	4.44 × 10^−1^	1.89 × 10^3^
	(5.27 × 10^−2^)	(5.47 × 10^−2^)	(3.90 × 10^−1^)	(2.52 × 10^−1^)	(8.87 × 10^1^)
F7	1.01 × 10^−3^	8.77 × 10^−4^	**6.64 × 10^−4^**	3.36 × 10^−3^	1.68 × 10^−3^
	(7.61 × 10^−4^)	(7.94 × 10^−4^)	(3.41 × 10^−4^)	(3.03 × 10^−3^)	(1.01 × 10^−3^)
F8	**−1.26 × 10^4^**	−1.25 × 10^4^	−8.57 × 10^3^	−1.05 × 10^3^	−7.53 × 10^3^
	(1.14)	(6.37 × 10^2^)	(6.76 × 10^2^)	(1.81 × 10^2^)	(9.28 × 10^2^)
F9	**0.00**	**0.00**	**0.00**	3.78 × 10^−15^	1.20
	(0.00	(0.00)	(0.00)	(1.44 × 10^−14^)	(2.37)
F10	**4.44 × 10^−16^**	**4.44 × 10^−16^**	**4.44 × 10^−16^**	3.16 × 10^−15^	7.31
	(0.00)	(0.00)	(0.00)	(2.58 × 10^−15^)	(5.43)
F11	**0.00**	**0.00**	**0.00**	3.36 × 10^−3^	4.63 × 10^−3^
	(0.00)	(0.00)	(0.00)	(7.25 × 10^−3^)	(2.53 × 10^−2^)
F12	**1.33 × 10^−4^**	6.86 × 10^−4^	5.03 × 10^−2^	2.84 × 10^−2^	4.01
	(6.77 × 10^−4^)	(2.41 × 10^−3^)	(2.45 × 10^−2^)	(4.33 × 10^−2^)	(1.93)
F13	**2.63 × 10^−1^**	7.50 × 10^−1^	6.43 × 10^−1^	5.30 × 10^−1^	1.23 × 10^1^
	(1.63 × 10^−1^)	(5.16 × 10^−1^)	(2.42 × 10^−1^)	(2.89 × 10^−1^)	(8.60)
F14	**9.98 × 10^−1^**	1.88	7.16	3.41	4.36
	(0.00)	(2.14)	(5.64)	(3.58)	(3.84)
F15	**3.07 × 10^−4^**	8.07 × 10^−4^	3.48 × 10^−4^	7.02 × 10^−3^	1.09 × 10^−3^
	(4.57 × 10^−7^)	(4.40 × 10^−4^)	(1.74 × 10^−4^)	(8.61 × 10^−3^)	(2.22 × 10^−3^)
F16	**−1.03**	**−1.03**	**−1.03**	**−1.03**	**−1.03**
	(1.43 × 10^−5^)	(6.11 × 10^−16^)	(5.13 × 10^−16^)	(4.81 × 10^−9^)	(2.79 × 10^−5^)
F17	**3.98 × 10^−1^**	**3.98 × 10^−1^**	**3.98 × 10^−1^**	**3.98 × 10^−1^**	**3.98 × 10^−1^**
	(6.48 × 10^−16^)	(1.80 × 10^−5^)	(1.23 × 10^−5^)	(1.71 × 10^−6^)	(1.45 × 10^−5^)
F18	**3.00**	**3.00**	**3.00**	**3.00**	**3.00**
	(1.18 × 10^−15^)	(2.44 × 10^−15^)	(1.01 × 10^−7^)	(1.35 × 10^−6^)	(1.13 × 10^−5^)
F19	**−3.85**	−3.81	−3.81	−3.81	−3.81
	(2.68 × 10^−15^)	(3.39 × 10^−3^)	(2.26 × 10^−15^)	(1.84 × 10^−2^)	(2.72 × 10^−3^)
F20	**−3.29**	−3.24	−3.27	−3.26	−3.18
	(1.49 × 10^−7^)	(7.53 × 10^−2^)	(5.99 × 10^−2^)	(9.17 × 10^−2^)	(1.53 × 10^−1^)
F21	**−8.89**	−8.46	−1.01 × 10^1^	−7.69	−9.61
	(2.36)	(2.42)	(4.89 × 10^−3^)	(2.92)	(1.54)
F22	**−8.11**	−1.03 × 10^1^	−1.02 × 10^1^	−8.08	−1.02 × 10^1^
	(6.60 × 10^−3^)	(2.65 × 10^−1^)	(9.62 × 10^−1^)	(2.88)	(5.06 × 10^−1^)
F23	**−8.67**	−1.05 × 10^1^	−1.01 × 10^1^	−7.75	−1.02 × 10^1^
	(7.73 × 10^−3^)	(2.55)	(1.74)	(3.10)	(1.57)

**Table 2 sensors-24-01734-t002:** Comparison of the performance of IDBO and the rival algorithms (F1–F23). NaN indicates that the results of the two algorithms were too similar to be significant. Values with *p*-values greater than 0.05 are represented by bolding.

F	IDBO vs. DBO	IDBO vs. SSA	IDBO vs. WOA	IDBO vs. PSO
F1	1.21 × 10^−12^	1.21 × 10^−12^	1.21 × 10^−12^	1.21 × 10^−12^
F2	1.21 × 10^−12^	1.21 × 10^−12^	1.21 × 10^−12^	1.21 × 10^−12^
F3	1.65 × 10^−9^	1.36 × 10^−11^	6.47 × 10^−12^	6.47 × 10^−12^
F4	3.54 × 10^−10^	1.96 × 10^−10^	6.47 × 10^−12^	6.47 × 10^−12^
F5	3.01 × 10^−11^	3.01 × 10^−11^	1.38 × 10^−6^	4.18 × 10^−9^
F6	3.01 × 10^−11^	3.01 × 10^−11^	3.01 × 10^−11^	3.01 × 10^−11^
F7	**5.60 × 10^−2^**	**8.70 × 10^−1^**	3.36 × 10^−6^	7.73 × 10^−6^
F8	1.85 × 10^−2^	3.01 × 10^−11^	3.01 × 10^−11^	3.01 × 10^−11^
F9	**NaN**	**NaN**	1.61 × 10^−2^	1.09 × 10^−12^
F10	**NaN**	**NaN**	7.20 × 10^−7^	1.20 × 10^−12^
F11	**NaN**	**NaN**	1.10 × 10^−2^	1.21 × 10^−12^
F12	1.95 × 10^−3^	3.01 × 10^−11^	3.01 × 10^−11^	3.01 × 10^−11^
F13	1.68 × 10^−4^	6.52 × 10^−8^	6.76 × 10^−5^	3.01 × 10^−11^
F14	1.24 × 10^−5^	5.69 × 10^−9^	1.21 × 10^−12^	1.21 × 10^−12^
F15	4.19 × 10^−10^	1.86 × 10^−6^	3.01 × 10^−11^	2.37 × 10^−10^
F16	1.13 × 10^−11^	7.57 × 10^−12^	3.01 × 10^−11^	2.28 × 10^−1^
F17	**NaN**	**NaN**	1.21 × 10^−12^	1.21 × 10^−12^
F18	5.50 × 10^−4^	3.60 × 10^−10^	1.77 × 10^−11^	1.77 × 10^−11^
F19	6.41 × 10^−5^	4.10 × 10^−12^	6.70 × 10^−3^	3.80 × 10^−7^
F20	2.00 × 10^−3^	5.10 × 10^−3^	1.09 × 10^−11^	1.09 × 10^−11^
F21	4.13 × 10^−3^	1.17 × 10^−2^	6.73 × 10^−6^	5.10 × 10^−3^
F22	4.13 × 10^−3^	1.02 × 10^−6^	8.29 × 10^−6^	8.12 × 10^−4^
F23	3.90 × 10^−2^	5.18 × 10^−7^	5.26 × 10^−4^	8.00 × 10^−3^

**Table 3 sensors-24-01734-t003:** The parameters optimized by the IDBO.

Parameters	$C_{0}$	$C_{1}$	d	σ	C
Optimal value	21.37	0.23	2	68.92	85.09

**Table 4 sensors-24-01734-t004:** Comparison of the predictive performance between IDBO-HKELM and the rival models.

Models	MAE	RMSE
BPNN	27.74	69.09
ELM	15.44	41.05
KELM	14.72	36. 93
IDBO-HKELM	8.33	20.64

## Data Availability

The data presented in this study are available on request from the corresponding author. The data are not publicly available due to the confidentiality requirements of the data provider.

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
