# Peer review of "Predictions of Aeroengines’ Infrared Radiation Characteristics Based on HKELM Optimized by the Improved Dung Beetle Optimizer"

_sensors, 2024, doi:10.3390/s24061734_

Round 1

Reviewer 1 Report

Comments and Suggestions for Authors

A novel approach is introduced to address the computational inefficiency inherent in simulating aero-engine exhaust systems. This method utilizes a hybrid kernel extreme learning machine (HKELM) optimized by an improved dung beetle optimizer (IDBO) for predicting infrared radiation characteristics. The IDBO algorithm integrates Levy flight and variable spiral strategies, enhancing the dung beetle optimizer's (DBO) optimization performance. Through rigorous testing on 23 benchmark functions, the IDBO algorithm demonstrates its superiority over other prominent metaheuristic algorithms, as confirmed by the Wilcoxon signed-rank test. Furthermore, the IDBO algorithm effectively optimizes the hyperparameters of HKELM. Applying the resulting IDBO-HKELM model to predict the infrared radiation characteristics of a typical axisymmetric nozzle yields promising results, with RMSE and MAE values of 20.64 and 8.83, respectively. These findings underscore the accuracy and feasibility of the proposed method for predicting aero-engine infrared radiation characteristics. 

The findings are accurate, and I believe they offer significant interest and importance within this context and are suitable for your journal.

 In conclusion, I recommend the paper for publication.

Author Response

Thank you very much for taking the time to review this manuscript. I am very grateful for your careful review and valuable suggestions on my manuscript. I am very pleased to learn that you have agreed to publish my article, which is a great encouragement to me. Thanks again for your support and help!

Reviewer 2 Report

Comments and Suggestions for Authors

Paper Overview:

1. The paper addresses a practical problem related to aeroengine exhaust system simulation calculation.

2. The use of a hybrid kernel extreme learning machine (HKELM) optimized by an improved dung beetle optimizer (IDBO) shows innovation in methodology.

3. The introduction of the Levy flight strategy and variable spiral strategy to enhance the optimization performance of the DBO algorithm demonstrates a novel approach.

4. The experimental validation of the IDBO algorithm against other metaheuristic algorithms using benchmark functions adds credibility to the proposed method.

5. The application of the IDBO-HKELM model for infrared radiation characteristics prediction in a typical axisymmetric nozzle provides concrete results that verify the accuracy and feasibility of the approach.

Required Revisions:

1. The paper could benefit from providing more detailed explanations of the methodologies used, particularly for readers who may not be familiar with HKELM or DBO algorithms.

2. The limitations and optimizer details of the proposed method and potential areas for future research could be discussed to provide a more comprehensive understanding of the work.

3. The paper could include a discussion on the practical implications of the results obtained, such as how this method could be applied in real-world aeroengine design or maintenance scenarios.

4. The paper could benefit from clearer presentation and comparison with other reported papers for the experimental setup and methodology used to validate the IDBO algorithm against other metaheuristic algorithms.

5. The paper could provide more insight into the generalizability of the proposed method to different types of aeroengine systems or scenarios.

6. The paper could benefit from a more detailed discussion on the computational complexity and efficiency of the IDBO-HKELM model. Understanding the trade-offs between accuracy and computational cost is crucial for practical implementation in real-world scenarios, especially in time-sensitive applications like aeroengine design and optimization.

7. The experimental validation of the IDBO algorithm against benchmark functions is essential, but the relevance of these functions to aeroengine exhaust system simulation may be limited. Including additional case studies or real-world examples related to aeroengine design would strengthen the credibility and applicability of the proposed method.

8. The paper could provide more insight into the robustness and reliability of the IDBO-HKELM model under different input parameters or environmental conditions. A thorough sensitivity analysis or robustness testing would enhance the confidence in the proposed approach and its potential for practical use in diverse aeroengine scenarios.

Author Response

Thank you very much for taking the time to review this manuscript. Please find the detailed responses below and the corresponding revisions/corrections highlighted/in track changes in the re-submitted files. Please see the attachment.

Reviewer 3 Report

Comments and Suggestions for Authors

The proposed method for predicting infrared radiation characteristics in the simulation calculation of aeroengine exhaust systems. The key components of the method include a hybrid kernel extreme learning machine (HKELM) and an optimization algorithm called the improved dung beetle optimizer (IDBO).     

Below are the comments.

1.       The introduction of the kernel matrix, defined as 𝛺𝐾𝐸𝐿𝑀(𝑿), is well-explained. However, it would be helpful to provide more context on why a kernel matrix is used in the context of the HKELM and how it contributes to the overall methodology.

2.       The derivation of the output matrix 𝑓𝐾𝐸𝐿𝑀(𝑿) using the pseudo-inverse matrix of 𝑯 is outlined clearly. However, providing additional information on the significance of this output matrix in the context of predicting infrared radiation characteristics could enhance understanding.

3.       The introduction of the regularized item in Formulas (5) and (6) is important for controlling the complexity of the model. However, the text could benefit from a brief explanation of the role of regularization in preventing overfitting and improving generalization.

4.       The text mentions that the hyper parameters of the hybrid kernel function can be optimized. Providing details on the optimization process, such as the objective function and optimization algorithm used, would offer a more complete understanding.

5.       The derivation of the hybrid kernel function in Formulas (10) and (11) is outlined, showcasing the combination of advantages from both global and local kernels. It would be beneficial to provide insights into how the hybrid kernel improves the model's performance in predicting infrared radiation characteristics.

6.       Equations (12) and (13) describe the position update for ball-rolling dung beetles encountering obstacles and the subsequent dance repositioning. The inclusion of parameters such as 𝛼, 𝑏, and 𝜃 is clear, but a brief explanation of their roles in the model would enhance understanding.

7.       Equations (14) describe the boundary selection strategy, simulating the regions where female beetles lay eggs. The inclusion of 𝑅 in the equation introduces a time-dependent aspect, but a brief explanation of its significance and how it influences the algorithm's behavior would be beneficial.

8.       Equations (15) and (17) provide the update rules for the brood ball and small dung beetle positions, respectively. The introduction of random vectors 𝑏1 and 𝑏2 adds a stochastic element to the movement. A brief explanation of how these updates contribute to the optimization process could enhance clarity.

9.       Equation (18) describes the position update for thieves mimicking the behavior of taking other dung beetles' dung balls. The introduction of parameters 𝑆 and 𝑔 is clear, but their roles in the theft behavior could be explained to provide a more comprehensive understanding.

10.   The claim that the IDBO-HKELM model outperforms other currently prominent machine learning methods needs more detailed validation. Including specific performance metrics, comparisons with other models, and potential limitations would strengthen the credibility of the findings.

11.   While the focus is on predictive performance, it is essential to consider the interpretability of the model, especially in aerospace applications where understanding the underlying physical principles is crucial. Discussing how well the ML models align with the known physics of infrared radiation in aircraft systems would add depth to the analysis.

12.   It is crucial to address the robustness and generalizability of the proposed model. How well does the IDBO-HKELM generalize to different scenarios or datasets? Robust validation on diverse datasets would enhance the reliability of the model.

13.   While the termination requirement is mentioned, it would be beneficial to provide details on how the algorithm determines convergence or when to stop the iterations.

Author Response

(The authors gave the same response as above.)

Round 2

Reviewer 2 Report

Comments and Suggestions for Authors

The author responses about my requested revisions are reasonable . 

Reviewer 3 Report

Comments and Suggestions for Authors

The suggested modifications are incorporated.